# Multiple Points Change in the Association of Blood Pressure Subtypes with Anthropometric Indices of Adiposity among Children in a Rural Population

**DOI:** 10.3390/children7040028

**Published:** 2020-04-02

**Authors:** Peter M. Mphekgwana, Kotsedi D. Monyeki, Herbert M. Makgopa, Phuti J. Makgae

**Affiliations:** 1Research Administration and Development, University of Limpopo, Polokwane, Sovenga 0727, South Africa; 2Department of Physiology and Environmental Health, University of Limpopo, Polokwane, Sovenga 0727, South Africa; kotsedi.monyeki@ul.ac.za (K.D.M.); phutimakgae@gmail.com (P.J.M.); 3Department of Pathology and Medical Sciences, University of Limpopo, Polokwane, Sovenga 0727, South Africa; Herbert.Mabalane@gmail.com

**Keywords:** anthropometric indices of adiposity, blood pressure, skinfold, hypertension

## Abstract

Background: Hypertension has gained global significance and risk of cardiovascular disease, and adiposity is the most important of the conditions associated with and considered responsible for hypertension in children. Therefore, the present study aimed to determine whether indices of adiposity independently predicted blood pressure at multiple points in gender-specific groups. Methods: This was a cross-sectional study involving 10 randomly selected primary schools within the Ellisras Longitudinal Study, and involved 1816 adolescents (876 girls and 940 boys) aged 8 to 17 years. All the anthropometric indices and blood pressures (BP) were examined according to the International Society for the Advancement of Kinanthropometry protocol. Results: In an adjusted linear quantile regression analysis of boys, waist circumference (WC) was associated with BP across all multiple points of systolic blood pressure (SBP). Furthermore, the triceps skinfold site was associated with high SBP. In girls, body mass index (BMI) was significantly associated with SBP after adjustment for potential confounders. Other anthropometric indices of adiposity, including WC, biceps, and triceps skinfold sites were not associated with SBP. Conclusions: The results of the present study suggest that in black South African children, variables such as WC and triceps skinfold site may provide stronger explanatory capacity to SBP variance and systolic hypertension risk in boys than other adiposity indices; whereas in girls, only WC and BMI predict diastolic blood pressure (DBP) and SBP, respectively.

## 1. Introduction

Hypertension is the most common risk factor for cardiovascular disease, the leading global cause of death [1]. Hypertension is a problem not only confined to developed countries as was previously reported [1,2,3], but is also increasing in developing countries undergoing an epidemiological transition from undeveloped to developed statuses, such as South Africa [2]. The burden of hypertension in South Africa is reported as one of the highest in Sub-Saharan Africa [3]. Between 42% to 54% of South Africans suffer from hypertension, and this figure is expected to increase exponentially if no urgent actions are taken [4]. 

Although the large component of hypertension occurs in adulthood, hypertension is considered a lifelong problem that can be progressive from childhood into adolescence and adulthood, such that individuals who have higher blood pressure in early life may have higher blood pressure values in later life [5]. Research on childhood hypertension has accumulated rapidly, and considerable progress has been made to identify factors associated with and are considered responsible for high blood pressure (BP) in children and adolescents [5]. Among these factors, indices of central (waist circumference), general (body mass index), or subcutaneous (skin-fold thickness) adiposity as the results of lifestyle, physical activity, and dietary habits are the most important of the conditions associated with elevated BP in childhood [6]. 

However, information on this association is still limited in the Limpopo Province of South Africa, especially concerning rural children and adolescents, as more emphasis has been placed on the middle aged and elderly population. It is, therefore, in this context that the present study aimed to determine, in black South African children aged 8 to 17 years in the Ellisras Longitudinal Study, whether indices of central (waist circumference), general (body mass index), or subcutaneous (skin-fold thickness) adiposity independently predicted BP at multiple points. This is particularly pertinent, given that Ellisras is reported to be dominated by cardiovascular risk factors, including among others, overweight or obesity and hypertension [7,8]. Researching this issue is of major interest to public health and has important policy implications in ensuring the health of young people today and adults of tomorrow. 

## 2. Materials and Methods

### 2.1. Study Design

This was a cross-sectional study involving 10 randomly selected primary schools within the Ellisras rural area. A total of 1816 children (876 girls and 940 boys) aged 8 to 17 years participated in the study. Informed consent was obtained from all participants, and study protocols were approved by the Turfloop Research Ethics Committee (TREC). Only children who were part of the Ellisras Longitudinal Study and returned the consent form were eligible to participate in the study. Children who did not submit the informed consent form were excluded from the study, together with those who were pregnant.

#### 2.1.1. Measurement

##### Anthropometric Measurements

All children underwent a series of anthropometric measurements according to the International Society for the Advancement of Kinanthropometry protocol [9]. Height was measured using a Martin Anthropometer to the nearest 0.1 cm, with the head in a Frankfort plane and the subject being in an anatomical position. Bodyweight was measured without shoes and with light clothing to the nearest 0.1 kg on electronic scales. Body mass index (BMI) was calculated by dividing weight by height in meter squared (kg/m^2^). Waist circumference (WC) was measured to the nearest 0.1 cm at the midway between the lowest rib and iliac crest with flexible steel tape [9]. Skinfolds thickness (triceps and biceps) was measured three times with a Slime guide caliber to the nearest 1 mm [9].

##### Blood Pressure (BP) Measurements

Blood pressure was measured using an electronic Micronta monitoring kit. At least three BP readings of systolic blood pressure (SBP) and diastolic blood pressure (DBP) were taken at an interval of five minutes apart, after the child had been seated for five minutes or longer [10,11]. The bladder of the device contains an electronic infrasonic transducer that monitors the BP and pulse rate, displaying these concurrently on the screen. This versatile instrument was designed for research and clinical purposes. In a pilot study conducted before the survey, a high correlation (*r* = 0.93) was found between the readings taken with the automated device and those taken with a conventional mercury sphygmomanometer [12].

##### Quality Control

All training of anthropometric measurements was done following the standard procedures of the International Society for the Advancement of Kinanthropometry (ISAK) [9]. The reliability and validity of anthropometric measurements were reported elsewhere [13,14]. The absolute and relative values for intra- and inter-tester technical error of measurements (% TEM) for height ranged from 0.04 to 4.16 cm (0.20%–5.01%), which was within the 5.1% acceptable rates, as reported by Norton and Olds [15].

### 2.2. Statistical Analyses

The parametric z-test was applied to test the significant differences between sexes at a 5% significance level. The parametric linear quantile regression approach was used in this cross-sectional study to examine the relationship between anthropometric indices and blood pressure subtype levels. The predictive variables included in the models, were age, waist circumference (WC), body mass index (BMI), triceps, and biceps skinfold sites. All the statistical analyses were performed using Stata 13 and R software.

## 3. Results

The characteristics of the participants are described in Table 1. Of the 1816 participants, 876 (48%) were girls and 940 (52%) were boys. The average age of girls and boys were 14 and 13 years, respectively, with no significant difference between the sexes as far as age was concerned (*p* = 0.548). There were significant differences between boys and girls with regard to SBP (*p* < 0.001), DBP (*p* < 0.001), BMI (*p* < 0.001), triceps skinfold site (*p* < 0.001), and biceps skinfold site (*p* < 0.001), but no significant differences between the sexes with regard to WC were found (*p* = 0.240). On average, girls had higher values for their SBP, BMI, and skinfold sites than boys did.

Table 2 illustrates all three quantile values of both SBP and DBP. The 25th, 50th, and 90th quantiles correspond with 99 mmHg (225 number of participants), 106 mmHg (226), and 118 mmHg (348) of SBP in boys and girls, respectively. The 20th, 50th, and 85th quantiles correspond with 61 mmHg (473), 67 mmHg, and 74 mmHg of DBP, respectively. The second quantile is 50 percentile, which reflects the median of all data.

Table 3 shows the linear quantile regression for the association of systolic blood pressure with anthropometric indices of adiposity. In boys, WC was associated with BP across all multiple points of SBP (*p* = 0,000 for low value of SBP, *p* = 0.001 for mid-SBP, and *p* = 0.009 for high value of SBP). Furthermore, the triceps skinfold site (*p* = 0.008) was significantly associated with high SBP (*p* = 0.008), whilst BMI and biceps skinfold site were not significantly associated with SBP in boys (all *p* > 0.05). In girls, BMI was significantly associated with mid-SBP (*p* = 0.009). Other anthropometric indices of adiposity, including WC, and biceps and triceps skinfold sites were not associated with SBP (all *p* > 0.05) in girls.

Table 4 shows a relationship between DBP and WC, BMI, triceps and biceps skinfolds sites in different quantiles for both girls and boys. It was found that WC was positively associated with low (*p* = 0.002) and high DBP (*p* = 0.031) in girls. Additionally, BMI, triceps and biceps skinfold sites were not significantly associated with outcome variable DBP after adjusting for confounder variable age (all *p* > 0.05).

## 4. Discussion

The present study presents a rare insight into rural-dwelling black South African children for multiple points change in the association of blood pressure subtypes with anthropometric indices of adiposity. The results revealed that there was a significant difference in the systolic and diastolic blood pressures of girls and boys. Girls had higher mean systolic and diastolic blood pressure than boys. This is inconsistent with the earlier findings [16], which showed lower systolic and diastolic blood pressure in girls than boys. The inconsistency between this study and that of Syme et al. may be due to differences in the characteristics of the study participants. For example, Syme et al. only included adolescents aged 12 to 18 years, in which case age may be likely to mask the levels of blood pressure.

The previous study by Al-Sendi et al., has found a strong association between anthropometric variables (WC, BMI, triceps skinfold, fat mass, weight, and height) with SBP in both genders [17]. However, the present study only found an association between WC and triceps skinfold site, except for BMI and biceps skinfold site. This is in agreement with the previous study by Emha et al., in which BMI was not associated with BP in hypertensive patients in Cardiology Polyclinic [18]. The reason for the lack of a significant association of BMI with SBP is thought to be the influence of potential confounders, such as diet and physical inactivity, which were not adjusted for [19,20]. Furthermore, DBP did not show any association with the adiposity indices, which affirm the results of the previous study [21], suggesting that SBP may be a better guide than DBP to evaluate cardiovascular and all mortality [22].

Consistent with the literature [23], the present study found that, in girls, BMI was positively associated with high SBP on adjusted linear quantile regression. The previous study by Ononamadu et al. reported BMI as one of the good predictors of the risk of development of both prehypertension and hypertension [24]. However, in the present study, no association was found between BMI and DBP.

The mechanism linking anthropometric measurements with low BP in children is not clear. However, it is thought that different anthropometric measurements may have varied impacts on BP. 

The results of the present study showed that only WC and BMI were significantly associated with low SBP in both boys and girls, whereas WC was only associated with low DBP in girls, which suggests anthropometric gender differences in this population. Several reasons could explain anthropometric gender differences in this population. Firstly, findings similar to those in this study were noted among children, with hormonal changes occurring at puberty being attributed to the observed differences [25]. Secondly, the psychosocial stress at menarche and rapid physiological changes that accelerate the completion of puberty among females were reported to be associated with an alteration in anthropometric measurements among female children [26].

Although the association between anthropometric indices of adiposity and BP are well established [6,18,21], the proposed mechanisms by which measures of adiposity directly affect BP is not clear. However, the study by Hall et al. stated that the mechanism between measurements of anthropometric indices of adiposity and BP could be considered as involving activation of the sympathetic nervous system (SNS); the amount of intra-abdominal and intra-vascular fat (as indicated by greater WC, WHR, and WHtR), rather than fat mass (as indicated by BMI); sodium retention, increase in renal reabsorption; and the renin-angiotensin system [27]. Accordingly, high amounts of intra-abdominal and intra-vascular fat are considered to be a chronic inflammation state, in which the excess accumulation of visceral adipose tissue (VAT) plays a central role [28]. VAT is a specific type of portal adipose tissue; it has high lipolytic activity and lower response to antilipolytic effect, resulting in a high rate of free fatty acids (FFA) production [29]. The high concentration of FFA in portal circulation further inhibits the hepatic clearance of insulin, resulting in the production of angiotensinogen, among other adipokines [21]. An imbalance in angiotensinogen leads to the activation of the renin-angiotensin system (RAS), causing vasoconstriction and reabsorption of sodium [30]. The constriction of blood vessels, in turn, increases blood pressure and eventually the development of high blood pressure.

### Study Limitations

The study was affected by the following possible limitations, among others. Firstly, the study was a cross-sectional design, preventing the assertion of a causal association of anthropometric indices of adiposity and blood pressure. Secondly, the sampling technique used may limit the external validity of the study, in that results cannot be generalized to the greater community outside of this group or the total Limpopo Province population. 

## 5. Conclusions

The results of the present study suggests, that in boys, variables such as WC and triceps skinfold site may provide a stronger explanatory capacity to SBP variance risk than other variables; whereas, in girls, only WC and BMI predict DBP and SBP, respectively. Since, individually, BMI and biceps skinfold site were not associated with SBP in boys, whereas in girls WC, biceps, and triceps skinfold sites were not associated with SBP, future research may be needed to build on the results of the present study to investigate the combined effects of anthropometric indices of adiposity on BP. In an area dominated by cardiovascular risk factors, including, among others, overweight or obese and hypertension, there is a need for screening and monitoring of children’s anthropometric indices and blood pressures regularly so that an effective intervention for the prevention and management of chronic diseases may be developed.

## Figures and Tables

**Table 1 children-07-00028-t001:** Characteristics of the participants by gender.

	Girls (*n* = 876 ) Mean (CI)	Boys (*n* = 940) Mean (CI)	*p* Value
Age (years)	13.48 (13.36; 13.60)	13.41 (13.29; 13.53)	0.548
SBP (mmHg)	107.92 (107.21; 108.64)	105.15 (104.48; 105.82)	<0.001
DBP (mmHg)	68.59 (68.08; 69.09)	66.99 (66.48; 67.51)	<0.001
WC (cm)	59.62 (59.29; 59.96)	59.18 (58.91; 59.44)	0.240
BMI (kg/m^2^)	16.76 (16.59; 16.94)	15.74 (15.62; 15.86)	<0.001
Triceps skinfold site (mm)	10.14 (9.87; 10.40)	7.42 (7.26; 7.58)	<0.001
Biceps skinfold site (mm)	6.14 (5.96; 6.31)	4.37 (4.26; 4.48)	<0.001

*p* set at 0.05 were obtained with parametric Z-test; *n* = number of participants; CI = confidence interval; SBP = systolic blood pressure; DBP = diastolic blood pressure; WC = waist circumference; BMI = body mass index.

**Table 2 children-07-00028-t002:** Data distribution per quantile of SBP and DBP.

	SBP Quantile %			DBP Quantile %		
	25 (*n*)	50 (*n*)	90 (*n*)	20 (*n*)	50 (*n*)	85 (*n*)
Girls	99 (225)	106 (226)	118 (348)	61 (179)	67 (259)	74 (316)
Boys	101 (237)	108 (230)	121 (340)	62 (173)	69 (302)	77 (266)

*n* = number of participants.

**Table 3 children-07-00028-t003:** Linear quantile regression for the association of systolic blood pressure with anthropometric indices of adiposity.

	Girls			Boys		
	*P* _25_	*P* _50_	*P* _90_	*P* _25_	*P* _50_	*P* _90_
WC (cm) *p* value	0.01 (−0.28;0.30) 0.964	0.11 (−0.21;0.43) 0.511	0.12 (−0.28;0.52) 0.555	**0.599** **(0.31;0.89)** 0.000	**0.56** **(0.23;0.89)** 0.001	**0.42** **(0.11;0.74)** 0.009
BMI (kg/m^2^) *p* value	**0.87** **(0.02;1.80)** 0.054	**1.01** **(0.25;1.77)** 0.009	**0.79** **(−0.11;1.69)** 0.084	−0.13 (−0.74;0.48) 0.684	0.32 (−0.47;1.11) 0.421	0.24 (−0.62;1.10) 0.584
Triceps skinfold site (mm) *p* value	−0.22 (−0.76;0.32) 0.425	0.09 (−0.64;0.28) 0.450	−0.49 (−1.14;0.17) 0.147	0.12 (−0.53;0.77) 0.717	−0.15 (−0.78;0.47) 0.628	**−0.48** **(−0.83;−0.13)** 0.008
Biceps skinfold site (mm) *p* value	0.42 (−0.50;1.34) 0.370	0.09 (−0.57;0.75) 0.791	0.47 (−0.36;1.31) 0.267	−0.26 (−1.05;0.54) 0.529	0.42 (−0.64;1.48) 0.442	**1.15** **(−0.13;2.31)** 0.053

*P =* percentile; Bolded = significant at 5% level.

**Table 4 children-07-00028-t004:** Linear quantile regression for the association of diastolic blood pressure with anthropometric indices of adiposity.

	Girls			Boys		
	*P* _20_	*P* _50_	*P* _85_	*P* _20_	*P* _50_	*P* _85_
WC (cm) *p* value	**0.3398** **(−0.13;0.55)** 0.002	0.1427 (−0.06;0.35) 0.167	**0.2523** **(0.02;0.48)** 0.031	0.1312 (−0.03; 0.29) 0.106	0.1080 (−0.09;0.31) 0.294	**0.1670** **(−0.01;0.35)** 0.068
BMI (kg/m^2^) *p* value	0.0116 (−0.42;0.44) 0.958	0.0781 (−0.39;0.55) 0.746	−0.1531 (−0.71;0.41) 0.592	0.3121 (−0.22;0.84) 0.248	0.1173 (−0.33; 0.56) 0.606	0.1234 (−0.25;0.51) 0.526
Triceps skinfold site (mm) *p* value	0.3317 (−0.21;0.87) 0.226	0.1781 (−0.14;0.50) 0.278	−0.2183 (−0.70;0.26) 0.373	−0.0354 (−0.34;0.27) 0.821	−0.0413 (−0.32;0.24) 0.770	0.1379 (−0.21;0.49) 0.441
Biceps skinfold site (mm) *p* value	−0.4067 (−1.28;0.46) 0.359	−0.1503 (−0.69;0.38) 0.582	0.2949 (−0.28; 0.87) 0.315	0.2061 (−0.21;0.62) 0.326	0.2843 (−0.16;0.73) 0.210	−0.0431 (−0.55;0.47) 0.868

*P =* percentile; Bolded = significant at 5% level.

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
