# Peer review of "Multiple Points Change in the Association of Blood Pressure Subtypes with Anthropometric Indices of Adiposity among Children in a Rural Population"

_children, 2020, doi:10.3390/children7040028_

Round 1

Reviewer 1 Report

Multiple Points Change in the Association of Blood Pressure Subtypes with Anthropometric Indices of Adiposity Among Children in a Rural Population.

In this cross-sectional study the authors want to study the association between different indices of adiposity (body mass index, waist circumference and skin fold thickness) with blood pressure traits in a sample of children in South Africa. Children from 10 primary schools were randomly selected among the Ellisras Longitudinal Study. The study involved 1816 children/adolescent aged from 8 to 17 years. Indices of adiposity were correlated with systolic and diastolic blood pressure and the analysis was performed separately from males and females.

Below some comments and suggestions are reported:

  • Abstract, page 1, line 22:

Waist circumference (WC) should be used in the same way in all the text.

  • Materials and Methods, page 2, line 66:

Is not clear which sample of the Ellisar Longitudinal Study is involved in this study since the reference [9] reports only the initial cross-sectional data about 684 boys and 652 girls aged 3-10 (“Monyeki, K.D., Van Lenthe, F.J. and Steyn, N.P., 1999. Obesity: does it occur in African children in a rural community in South Africa?” International journal of epidemiology, 28(2), pp.287-292”).

In another cited work [7] (“Monyeki, K.D., Kemper, H.C.G. and Makgae, P.J., 2006. The association of fat patterning with blood pressure in rural South African children: the Ellisras Longitudinal Growth and Health Study”. International journal of epidemiology, 35(1), pp.114-120.) 1816 participants are reported but the age of the subjects ranged 7-13 years and a difference number of boys and girls.

  • Materials and Methods, page 2, line 72:

in the abstract the authors wrote that the anthropometric measurements were recorded accordingly to WHO STEPS, but here they refer to the International Society for the Advancement of Kinanthropometry protocol.

  • Materials and Methods, page 2, line 76:

For children, BMI should be transformed to percentiles based on reference tables for age and sex (see Extended international (IOTF) body mass index cut-offs for thinness, overweight and obesity, Cole et al 2012, Pediatr Obes). The authors are invited to provide these data, otherwise is difficult also to understand the prevalence of normal, excess weight and obesity of the population.

  • Materials and Methods, page 2, line 77:

the reference to the method of measurement of anthropometric parameter is missing.

  • Materials and Methods, page 2, paragraph Blood pressure measurements, line 87:

the reference to the pilot study the authors refers to, is missing.

  • Materials and Methods, page 2, paragraph Blood pressure measurements:

Since the subjects are children the authors are invited to compute percentiles for age and sex for systolic and diastolic blood pressure, accordingly to the “National Heart, Lung, and Blood Institute, 1996. Update on the 1987 Task Force Report on High Blood 245 Pressure in Children and Adolescents: A Work Group Report from the National High Blood Pressure Education Program. Pediatrics, 98, pp.649-657”. Percentiles are necessary to identify children with high blood pressure or hypertensive.

  • Results, paragraph characteristics of participants:

Information about prevalence of overweight/obesity and high blood pressure/hypertension is missing.

  • Discussion, line 138-139.

The reference [13], concerning sex differences in coronary heart disease and stroke is not relevant to the fact that the author found higher systolic blood pressure in girls.

  • Discussion, line 144 the authors wrote: “In agreement with studies by Al-Sendi et al., and Emha et al., the present study showed that in boys, WC and triceps skinfold site except for BMI and biceps skinfold site were associated with SBP on adjusted linear regression [14-15]”

The results by Al-Sendi and colleagues [14] and Ehma and colleagues [15] are not in line with the results obtained by the authors:  Both found a strong association with all the examined anthropometric variables with SBP, included BMI, and in both sexes. Moreover, subjects in Ehma and colleagues work are hypertensive patients in cardiology clinic.

The authors should provide more consistent references to support their results.

  • Discussion, line 144-146: “The reason for the lack of significant association of BMI with SBP is thought to be the influence of potential confounders such as diet and physical inactivity, which were not adjusted for.”:

The authors should provide a reference for this statement.

  • Discussion, line 146-148: Reference [17] is the duplicate of [7]

  • Discussion, line 146-148: “Furthermore, DBP did not show any association with the adiposity indices which affirm the results of the previous studies [16-17]”

In the study “Monyeki, K.D., Kemper, H.C.G. and Makgae, P.J., 2006. The association of fat patterning with blood 235 pressure in rural South African children: the Ellisras Longitudinal Growth and Health Study”. International journal of epidemiology, 35(1), pp.114-120”, in table 4, show significant association between DBP and anthropometric indices.

  • Discussion, line 148: “suggesting that SBP may be as a better guide 147 than DBP to evaluate cardiovascular and all mortality”.

The authors should provide a reference.

  • Discussion, line 150: “...Suggesting that BMI may be a good predictor of the risk of development of both hypotension and hypertension in girls.”

This cannot be inferred by the results presented by the authors.

  • Discussion, line 151-153: “However, no association was found between BMI and DBP, suggesting that the association of BMI with SBP and DBP may go in opposite directions. The fact that BMI is associated with BP cannot be disputed; however, in the study, BMI may be too low to have a large impact on DBP.

The results obtained by the authors don’t seem to suggest an opposite direction of DBP respect to SBP. Moreover, it is not clear why they wrote that the association between BMI and SBP cannot be disputed, and is not clear what they intended for BMI too low.

  • Conclusion, line 195:

The association to systolic hypertension is not presented in the study.

Author Response

Respond

Reviewer 2 Report

This study addressed anthropometric indices among children in a rural area from South Africa. The results are straightforward. Nevertheless, I have a few remarks.

  1. There is no clear hypothesis for this study; should be formulated.

  1. The statistical paragraph starts with a remark on differences between sexes; however, the authors fail to state why a test between sexes was necessary; again, one misses a hypothesis here. Significance testing is against a zero hypothesis, isn’t it?

  1. Given the age range, it would be better to break the groups up into prepubertal and pubertal groups and also take into account the distinction between girls at premenarchal age and those who have menstrual cycles. At which phase of the menstrual cycle were measurements obtained?

  1. More details should be provided regarding the blood pressure measurement. Has the device been validated according to international protocols and, if so, where is the reference? Is it oscillometric?

  1. The discussion should stick to the data; extrapolations to dizziness etc are meaningless.

  1. In itself, the data are not very novel. A better description in how these data fill the gaps in our knowledge is necessary.

Author Response

Respond

Round 2

Reviewer 1 Report

The authors improved the manuscript answering all the questions in a clear way. The actual version of the manuscript appear clear and consistent.

Reviewer 2 Report

The authors have not taken my comments seriously. Their answers to my criticisms are not true answers and they have not materially modified the manuscript.